# A Coupled Finite-Boundary Element Method for Efficient Dynamic Structure-Soil-Structure Interaction Modeling

Parham Azhir [1], Jafar Asgari Marnani [1,*], Mehdi Panji [2,*] and Mohammad Sadegh Rohanimanesh [1]

1 Department of Civil Engineering, Faculty of Civil and Earth Resources Engineering, Central Tehran Branch, Islamic Azad University, Tehran 1955847781, Iran; p.azhir.eng@iauctb.ac.ir (P.A.); m.s.rohanimanesh@iauctb.ac.ir (M.S.R.)
2 Department of Civil Engineering, Zanjan Branch, Islamic Azad University, Zanjan 5814545156, Iran
* Correspondence: ja.asgari@iau.ac.ir (J.A.M.); m.panji@iauz.ac.ir (M.P.)

**Abstract:** This paper introduces an innovative approach to numerically model Structure–Soil-Structure Interaction (SSSI) by integrating the Boundary Element Method (BEM) and the Finite Element Method (FEM) in a coupled manner. To assess the accuracy of the proposed method, a comparative study is undertaken, comparing its outcomes with those generated by the conventional FEM technique. Alongside accuracy, the computational efficiency aspect is crucial for the analysis of large-scale SSSI problems. Hence, the computational performance of the coupled BEM–FEM method undergoes a thorough examination and is compared with that of the standalone FEM method. The results from these comparisons illustrate the superior capabilities of the proposed method in comparison to the FEM method. The novel approach provides more reliable results compared to traditional FEM methods, serving as a valuable tool for engineers and researchers involved in structural analysis and design.

**Keywords:** structure-soil-structure interaction; boundary element method; finite element method; numerical modeling; computational efficiency





## 1. Introduction

Soil engineers place significant emphasis on understanding the load-bearing capacity of soil. In a notable example, Ajdari and Esmail Pour [1] conducted a comprehensive study, devising a specialized bearing capacity device to analyze the load-settlement behavior of circular footings situated above the groundwater table. Their experimental findings revealed that conventional equations used to estimate footing bearing capacity tend to be overly conservative. As a result, the authors proposed a specific empirical relationship tailored for circular footings. Additionally, extensive research efforts are dedicated to investigating the performance of footings on various soil types. Notably, Veiskarami and Kumar [2] presented a pragmatic approach for assessing the bearing capacity of surface footings on non-associative sand, validating this procedure through comparisons with data obtained from footing load tests.

On a different note, site engineers are generally more interested in the behavior of the free ground or the site in the existence of structures. For example, Kazemeini and Haghshenas [3] studied the impact of underground cavities on site seismic response in Karaj city, with a particular focus on areas adjacent to under-construction subway tunnels. Their research employed ambient noise measurements and numerical modeling at 11 test sites to evaluate the effect of the tunnel on seismic site response, revealing variations based on tunnel dimensions and proximity.

Soil–Structure Interaction (SSI) refers to the interplay between the soil and the structure, where both are considered as a single system [4]. The load from the structure is transferred to the soil, which responds by deforming or settling, potentially affecting the behavior of the structure and its response to external loads. SSI is a crucial aspect of geotechnical

and structural engineering as it significantly impacts the economy of construction, safety, and performance of structures. The study of SSI is a prominent research area for several decades [5–8], primarily motivated by the need to comprehend and predict the behavior of structures under different loading conditions. Initial investigations on SSI focused on analyzing shallow foundations' reaction to static loads using analytical techniques such as elastic half-space theory and the Cone method to simulate soil response and evaluate deformation and stress distribution beneath the foundation [9].

In the realm of structural engineering, the concept of SSI has garnered considerable attention due to its impact on the behavior and performance of structures in relation to the surrounding soil. However, the role of Structure–Soil-Structure Interaction (SSSI) in determining such behavior and performance under various loading conditions has received comparatively less attention. SSSI is a phenomenon that occurs when multiple structures are constructed on a common foundation, and each structure interacts with both the soil and other structures through their shared foundation. The behavior of each structure can affect its neighboring structures, leading to complex interactions between them. SSSI is an important consideration in the design and construction of buildings, bridges, and other infrastructure projects, as it can significantly impact their performance and safety. The dynamic behavior of soil and its interaction with structures and other surrounding structures strongly influences how structures respond to external loads, including earthquakes, wind, and traffic. Therefore, precise modeling of SSSI is essential for designing and analyzing structures, particularly in densely populated urban and industrial areas.

Recent advancements in SSSI analysis have seen significant progress, with researchers developing new methods and models to understand the behavior of multiple structures interacting with soil [10–14]. Using Finite Element Analysis (FEA) is one such method, considering factors like soil stiffness, foundation shape, and loading conditions [11]. Other researchers focus on accurate soil models, including nonlinear ones, to capture soft soil behavior and effects of soil liquefaction [13]. The primary obstacle hindering widespread application of the Finite Element Method (FEM) in SSI is suboptimal computational efficiency. Addressing this challenge, scholars propose innovative methodologies in various structural analysis aspects. For instance, Yu and Li [15] present a two-stage method for efficient nonlinear seismic response analysis, and Li and Yu [16] introduce the Inelasticity-Separated Finite-Element Method (IS-FEM), enhancing efficiency by decomposing strain and applying a dynamic stiffness matrix. Simpson and Zhu [17] explore GPU acceleration, showing improved performance. These endeavors collectively aim to overcome computational inefficiencies, advancing the broader utility of the finite element method in effectively addressing challenges in soil–structure interaction.

Recent studies have also investigated the impact of SSSI on different types of structures, including high-rise buildings, bridges, and tunnels [10,14]. These studies have highlighted the need for accurate modeling and analysis of SSSI to ensure the safety and reliability of these structures. For example, researchers have used FEA to analyze the effects of SSSI and SSI on tall buildings during earthquakes and fount that adjacent buildings can significantly affect each other's behavior due to soil movement and deformation [10]. Bariker and Kolathayar [18] have employed the FEA method to analyze a new foundation construction. Other studies have looked at the effects of SSSI on underground structures such as tunnels, finding that soil–structure interaction can cause significant stress and deformation in the tunnel [19,20]. Overall, the state of the art in SSSI analysis is rapidly advancing, with researchers developing new methods and models to better understand the complex interactions between structures and soil.

The coupled FEM-Boundary Element Method (BEM) technique is initially introduced in a publication authored by Zienkiewicz and Kelly [21] in 1977. Furthermore, a comprehensive review of the relevant literature concerning this subject can be found in the work of Hong-Bao and Guo-Ming [22] in 1986. In recent years, there has been a growing interest in the development and analysis of novel methods for coupling FEM–BEM. Extensive research has been conducted in various fields such as fluid and solid mechanics, geomechanics,

electromagnetics, and acoustics [23–27]. Existing approaches for coupling can be broadly classified into three categories: FEM-hosted, BEM-hosted, and those that do not fall into either of these two categories.

The first category involves treating the subdomain governed by the BEM method as a macro-finite element, or super-element. This entails converting the displacement traction equations that govern the boundary element subdomain into displacement force equations, which are then combined with those of FEM. This approach has been explored by researchers in studies [27–30].

Conversely, the BEM approach treats the Finite Element (FE) subdomain as an equivalent subregion governed with BEM-like equations. This involves converting the stiffness equations of the FEM to equations resembling those of the BEM, while ensuring continuity and equilibrium along the interface. This methodology was originally proposed by Zienkiewicz and Kelly [21] and further discussed by [31] in 1979.

Approaches that do not fall within either of these two categories involve direct coupling. However, such approaches are challenging and inefficient due to the substantial number of unknowns involved, as pointed out by Ganguly and Layton [30] in 2000. One such approach is the boundary coupling method proposed by Hsiao [32] in 1988, where the governing equations of one subdomain are treated as the boundary conditions for the other subdomain. In a similar vein, an alternative method called iterative domain decomposition coupling has been developed by Lin and Lawton [33] in 1996 and enhanced by Elleithy and Al-Gahtani [34] in 2001. This iterative approach solves the original problem by continuously adjusting the unbalanced forces or displacements from the subdomains to the artificial interface until continuity and equilibrium conditions are satisfied. However, one drawback of this technique is that it requires solving the boundary problems in both the BEM and FEM subregions at each iteration step. Due to the potentially slow convergence of the process, this can lead to long computational times.

This paper presents a new approach to SSSI analysis that combines the FEM and BEM based on the method proposed by Aour and Rahmani [29]. By leveraging the strengths of both methods, the proposed approach offers improved accuracy and computational efficiency. The effectiveness of this method is demonstrated through a case study of structures, where the behavior of the structure is accurately predicted using the proposed approach. Furthermore, the computational effort required by the proposed approach is significantly reduced compared to traditional methods. Overall, the results highlight the benefits of the novel FEM–BEM coupled approach in accurately analyzing SSSI while achieving computational efficiency, making it a valuable tool for engineers and researchers in the field of structural analysis and design.

## 2. Finite Element Method (FEM)

FEM is a fundamental numerical technique that has emerged as a crucial method in modern computational science. FEM is employed for solving partial differential equations by dividing complex systems into smaller, simpler parts known as finite elements. Each element is characterized by a set of equations that establish its behavior in relation to its neighboring elements. The system as a whole is then solved by assembling these equations into a large matrix and employing numerical methods. FEM finds extensive application in engineering and physics to analyze the intricate behavior of complex systems, encompassing areas such as structures, fluids, and electromagnetic fields.

Despite FEM popularity and wide applications in diverse fields of engineering, FEM suffers from a few drawbacks. Firstly, the accuracy of the results can be affected by the choice of the element size and mesh density, which can lead to errors in the solution. Secondly, the computational cost of FEM can be high, especially for large-scale problems. This can be attributed to the need for significant computational resources and the time-consuming pre-processing stage for generating the mesh. These challenges make it difficult to use FEM for certain applications. As noted by Bathe [35], accuracy and computer time

often conflict in FEM simulations. Additionally, Hughes [36] highlights the difficulties in FEM analysis for problems with complex geometries and material behavior.

### 2.1. Transient Finite Element Method (TFEM)

Within the FEM method, the resolution of complex boundary value problems entails the subdivision of the problem into discrete and solvable elements. The subsequent re-assembly of these elements culminates in the comprehensive solution to the boundary value problem. Dynamic FEM analyses are commonly classified into three distinct types: Natural Frequency Analysis (NFA), Harmonic Analysis (HA), and Transient Analysis (TA). NFA is specifically focused on elucidating the natural harmonic response of a structure, while HA scrutinizes the system's behavior across repeated time intervals. In the context of TA, time-varying conditions are applied to a system structure, and its response is extracted.

In the realm of FEM, Transient Analysis (TA) undergoes further categorization into Implicit and Explicit analyses, with due consideration given to diverse formulations [36]. For a system characterized by multiple degrees of freedom, the equations of motion can be succinctly expressed as outlined in references [36,37]:

$$[m]\{\ddot{d}\} + [c]\{\dot{d}\} + [k]\{d\} = \{f\} \tag{1}$$

where $\{d\}$ represents the displacement vector of the multi-degree of freedom system, and the matrices of $[m]$, $[c]$, and $[k]$ represent the mass, damping, and stiffness matrices of the system, respectively. The vector $\{f\}$ represents the external force vector of the system. It should be noted that the displacement vector $\{d\}$ is a function of time, denoted as $\{d\} = \{d(t)\}$, and upper dots indicate derivatives of displacement with respect to time. Therefore, $\{\ddot{d}\}$ represents the acceleration vector and $\{\dot{d}\}$ epresents the velocity vectors.

The stiffness matrix $[k]$ is initially defined as the static stiffness matrix, with subsequent definition and assembly of element stiffness matrices contributing to the overall stiffness matrix while maintaining system continuity. Simultaneously, matrices of mass $[m]$ and damping $[c]$ are derived. Within the TFEM method, displacement at a given time step is expressed using information from previous steps and the current time. Figure 1 depicts this predictive process, determining displacement at $d_{n+1}$ as an unknown value by integrating current $d_n$ and previous $d_{n-1}$ displacement information. Two primary formulations, explicit and implicit, govern this process, to be detailed later, with Table 1 offering a comprehensive comparison of their main specifications [37].

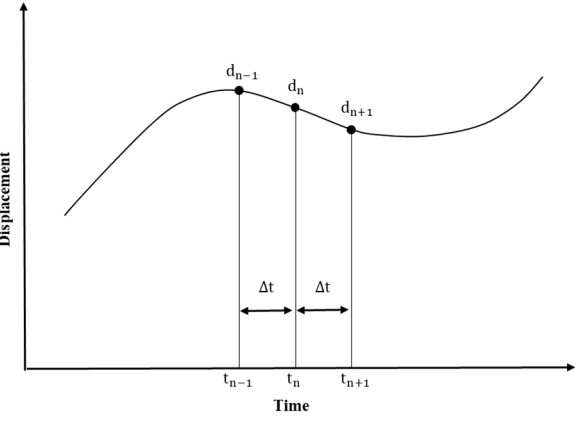

d_{n-1} − displacement at the previous time increment
d_n − displacement at the current time increment
d_{n+1} − displacement at the next time increments (unknown)

**Figure 1.** Displacement versus time in an arbitrary transient displacement.

**Table 1.** Comparison of the explicit and implicit methods.

|  | Implicit Transient Finite Element Method | Explicit Transient Finite Element Method |
|---|---|---|
| Matrix Inversion | Required | Not required (in case of Nilpotent damping matrix) |
| Time Integration | Euler | Central difference method |
| Stability | Stable for all time increments (precision improves with smaller time increments.) | Stable for relatively small-time increments (accurate with stable time increments) |

2.1.1. Implicit Transient Finite Element Method (ITFEM)

In this section, a brief explanation of the ITFEM is provided. Considering Equation (1) as the equation of motion, which is valid for all time steps, the equation of motion for the unknown time step, $n + 1$, can be expressed as follows:

$$[m]\left\{\ddot{d}_{n+1}\right\} + [c]\left\{\dot{d}_{n+1}\right\} + [k]\{d_{n+1}\} = \{f_{n+1}\} \tag{2}$$

Using Euler's formulation, the values of velocity and acceleration at time step $n + 1$ are established. It is important to note that by accounting for the variation in acceleration during a time increment, the accuracy of the method is enhanced [37], as defined by the values of $\alpha$ and $\beta$. Various values for $\alpha$ and $\beta$ are suggested in the literature. In this paper, we adopt conventional values of $\alpha = 0.5$ and $\beta = 0.25$. After replacing the values of velocity and acceleration, Equation (3) can be expressed as follows:

$$\begin{aligned}
\left(\frac{1}{\beta\Delta t^2}[m] + \frac{\alpha}{\beta\Delta t}[c] + [k]\right)&\{d\}_{n+1} \\
&= \{f\}_{n+1} + \left(\frac{1}{\beta\Delta t^2}[m] + \frac{\alpha}{\beta\Delta t}[c]\right)\{d\}_n \\
&\quad - \left(\frac{1}{\beta\Delta t}[m] + \left(\frac{\alpha}{\beta} - 1\right)[c]\right)\{\dot{d}\}_n \\
&\quad + \left(\left(\frac{1}{2\beta} - 1\right)[m] + \Delta t\left(\frac{\alpha}{2\beta} - 1\right)[c]\right)\{\ddot{d}\}_n
\end{aligned} \tag{3}$$

$$\left[k_{eff}\right]\{d\}_{n+1} = \left\{f_{eff}\right\}_{n+1} \tag{4}$$

Within this context, it is imperative to emphasize that the solution procedure mandates the inversion of the matrix $\left[k_{eff}\right]$, introducing a potential for relative time-consuming computations in the method. Nevertheless, a notable merit of this approach lies in its inherent stability across all time increments. This pivotal characteristic underscores the rationale for the implementation of ITFEM in the proposed methodology.

2.1.2. Explicit Transient Finite Element Method (ETFEM)

In this section, the ETFEM method is briefly explained. For time step $n$, Equation (1) yields as follows:

$$m\ddot{d}_n + c\dot{d}_n + kd_n = f_n \tag{5}$$

In the ETFEM, the velocity and acceleration at the current time step are expressed in relation to displacements at the preceding and subsequent time steps (the unknown time step). Subsequently, the equations are rearranged to facilitate the solution for $d_{n+1}$. Employing the central difference method to achieve this objective, the form of Equation (5) can be represented in matrix form as follows:

$$\left([m]\frac{1}{\Delta t^2} + [c]\frac{1}{2\Delta t}\right)\{d\}_{n+1}$$
$$= \{f\}_n + \left([m]\frac{2}{\Delta t^2} - [k]\right)\{d\}_n \tag{6}$$
$$- \left([m]\frac{1}{\Delta t^2} - [c]\frac{1}{2\Delta t}\right)\{d\}_{n-1}$$

In this methodology, it is imperative to posit a hypothetical value for the displacement at the time step preceding the initial displacement. This assumption is established according to the Euler formula and expressed in matrix form as follows:

$$\{d\}_{-1} = \{d\}_0 + \Delta t\{\dot{d}\}_0 + \left(\frac{1}{2}\right)\Delta t^2\{\ddot{d}\}_0 \tag{7}$$

In this context, it is noteworthy that when utilizing a lumped mass matrix and neglecting damping, matrix inversion becomes unnecessary, leading to the categorization of this method as "Explicit". Consequently, the computational effort required for each time step is significantly reduced. Another salient feature of this method is the imperative selection of a very small time step to ensure a stable solution for the problem. This characteristic renders the explicit method particularly suitable for analyzing short-duration phenomena, such as forging and explosions. However, it is essential to highlight that, for the proposed method in this paper, it exhibits certain limitations in comparison to the ITFEM method.

## 3. Boundary Element Method (BEM)

The Boundary Element Method is a numerical technique that is frequently used to solve partial differential equations over arbitrary geometries. According to Wrobel and Aliabadi [38], BEM applies various equations such as elastodynamic, Laplace, Helmholtz, or Poisson equations to solve problems in engineering, physics, and applied mathematics. One of the key advantages of BEM is its ability to eliminate the need for discretizing the entire domain, thus reducing the computational complexity of the problem. Furthermore, BEM offers an accurate and efficient approach for modeling problems with complex geometries or boundary conditions. This makes it a valuable tool for a wide range of applications [39].

The formulation of the Elastodynamic equation in the time domain for solving the system of equations at all boundary nodes is expressed in matrix form. The system of equations is represented as the sum of the product of matrix $H$ and vector $u$, which is equal to the sum of the product of matrix $G$ and vector $p$. In this context, the variables $u$ and $p$ represent displacement and traction in two distinct directions, respectively [39].

$$\sum_{m=1}^{n} H^{nm} u^m = \sum_{m=1}^{n} G^{nm} p^m \tag{8}$$

In the initial time step, Equation (8) can be deduced as $H^{11} * u^1 = G^{11} * p^1$. Similarly, in the subsequent time step, an additional equation is augmented to the equation from the previous time step, resulting in the following equation:

$$H^{21} u^1 + H^{22} u^2 = G^{21} p^1 + G^{22} p^2 \tag{9}$$

The aforementioned can be expressed as follows:

$$H^{21} u^1 = G^{22} p^2 + \left(G^{21} p^1 - H^{21} u^1\right) \tag{10}$$

At the third time step, the equation can be obtained by adding the third equation to the equation from the preceding time step. This mathematical operation results in a new equation that captures the changes between these two-time steps. By iteratively applying

this method, a series of equations can be generated that describe the evolution of the system over time, as follows:

$$H^{31}u^1 + H^{32}u^2 + H^{33}u^3 = G^{31}p^1 + G^{32}p^2 + G^{33}p^3 \tag{11}$$

The formula can be organized as follows:

$$H^{33}u^3 = G^{33}p^3 + Z^3 \tag{12}$$

$$Z^3 = G^{31}p^1 + G^{32}p^2 - H^{31}u^1 - H^{32}u^2 \tag{13}$$

It is important to acknowledge that the values of $H$ and $G$ depend on the difference between $n$ and $m$, rather than the specific values of $n$ and $m$ themselves. As a result, the equations can be reformulated as follows:

$$H^{11}u^3 = G^{11}p^3 + \left( G^{31}p^1 + G^{21}p^2 - H^{31}u^1 - H^{21}u^2 \right) \tag{14}$$

## 4. Hybrid Finite Element/Boundary Element Method (FEM/BEM)

The coupling of the finite element method and the boundary element method has become a topic of great interest in the field of computational mechanics. This coupling is particularly useful when dealing with problems involving unbounded domains, as it allows for the accurate modeling of both the finite and semi finite objects. According to recent research conducted by Gwinner and Stephan [40], the coupled finite-element–boundary-element method has shown to be successful in solving various engineering problems, such as acoustic radiation and fluid–structure interaction. The combination of these two methods provides significant advantages over the use of either method, and has the potential to revolutionize the analysis of complex engineering systems.

BEM exhibits limitations when analyzing behaviors within discrete domains; however, it demonstrates notable proficiency in analyzing infinite and semi-infinite domain behaviors. By integrating the FEM method with BEM, it becomes possible to preserve the advantages of both methods while alleviating their respective drawbacks. The coupling of FEM and BEM can be achieved through two primary schemes: BEM-hosted and FEM-hosted. In the BEM-hosted scheme, equations derived from the FEM formulation are converted into BEM equations. Conversely, in the more extensively explored FEM-hosted scheme, the BEM formulation is converted into the FEM formulation. This study is specifically dedicated to the implementation of the latter approach, known as the FEM-hosted methodology.

*FEM-Hosted Coupling of FEM and BEM*

In the FEM-hosted approach, the equations utilized in both FEM and BEM are transformed to align with the structure of FEM equations. This transformation process is illustrated in Figure 2.

While the ultimate sets of equations resulting from the FEM and BEM approaches may initially exhibit disparities, it is possible to manipulate them into a unified formulation. Specifically, for two subdomains, these transformed equations can be expressed as follows:

$$[K]^{FE}\{u\}^{FE} = \{F\}^{FE} \tag{15}$$

$$[H]^{BE}\{u\}^{BE} = [G]^{BE}\{t\}^{BE} \tag{16}$$

where $[K]^{FE}$ represents the stiffness matrix for the finite element subdomain, $\{u\}^{FE}$ and $\{F\}^{FE}$ denote the nodal displacement and force vectors, respectively. Similarly, $[H]^{BE}$ and $[G]^{BE}$ represent the influence coefficient matrices, while $\{u\}^{BE}$ and $\{t\}^{BE}$ represent the displacement and traction vectors of the boundary element subdomain.

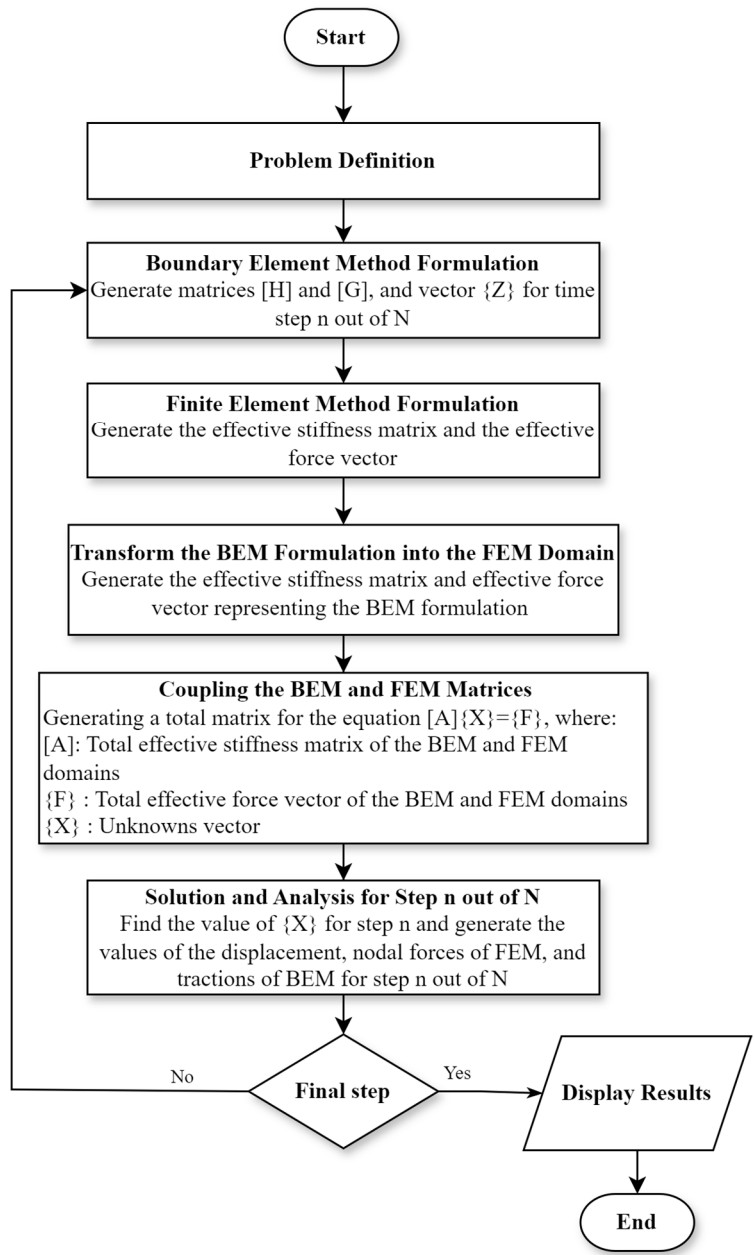

**Figure 2.** Schematic expression of FEM-hosted BEM–FEM coupling approach.

In the initial step, the nodal forces and the equivalent stiffness matrix of the boundary element subdomain are determined. Applying the virtual work principle, the nodal work of the forces on the boundary can be expressed as follows:

$$\delta W^e = (\delta u^e)^T F^e \tag{17}$$

where $\delta u^e$ represents the nodal displacement, $F^e$ denotes the nodal force, and $\delta W^e$ signifies the work done by the applied tractions, and is considered as follows:

$$\delta W^e = \int_\Gamma \left( t_x \delta u + t_y \delta v \right) d\Gamma \tag{18}$$

where $t_x$ and $t_y$ represent the tractions in the $x$ and $y$ directions, respectively, and $\delta u$ and $\delta v$ denote the displacements in the $x$ and $y$ directions, respectively. If the traction and

displacements are distributed on the boundary, taking into account the shape functions as follows:

$$\delta u(\xi) = \sum_{i=1}^{3} N_i(\xi).\delta u_i^e, \quad \delta v(\xi) = \sum_{i=1}^{3} N_i(\xi).\delta v_i^e \tag{19}$$

$$t_x(\xi) = \sum_{j=1}^{3} N_j(\xi).(t_x^e)_j, \quad t_y(\xi) = \sum_{j=1}^{3} N_j(\xi).\left(t_y^e\right)_j \tag{20}$$

where $N_i(\xi)$ is the shape function, and $\delta u_i^e$, $\delta v_i^e$ are the horizontal and vertical nodal displacement vectors. Furthermore, $t_x^e$, $t_y^e$ are the nodal traction vectors in the $x$ and $y$ directions, respectively. The work of the applied tractions can be expressed in the following form:

$$\delta W = \sum_{i=1}^{3} [\delta u_i \sum_{j=1}^{3} \left\{ (t_x)_i \int_\Gamma N_i(\xi)N_j(\xi)d\Gamma \right\} \\ + \sum_{j=1}^{3} \left\{ (t_y)_i \int_\Gamma N_i(\xi)N_j(\xi)d\Gamma \right\} \delta v_i] \tag{21}$$

The corresponding work done by the equivalent nodal force can be formulated as follows:

$$\delta W = \sum_{i=1}^{3} \left[ (F_x)_i \delta u_i + (F_y)_i \delta v_i \right] \tag{22}$$

$$(F_x)_i = \sum_{j=1}^{3} (t_x)_i \int_\Gamma N_i(\xi)N_j(\xi)d\Gamma \tag{23}$$

$$(F_y)_i = \sum_{j=1}^{3} (t_y)_i \int_\Gamma N_i(\xi)N_j(\xi)d\Gamma \tag{24}$$

or in matrix form, it can be written as follows:

$$F^e = M^e t^e \tag{25}$$

where $M^e$ is the converting matrix, which depends on the interpolation functions as follows:

$$M^e = \begin{bmatrix} \int_{-1}^{+1} N_1 N_1 J d\xi & 0 & \int_{-1}^{+1} N_1 N_2 J d\xi & 0 & \int_{-1}^{+1} N_1 N_3 J d\xi & 0 \\ 0 & \int_{-1}^{+1} N_1 N_1 J d\xi & 0 & \int_{-1}^{+1} N_1 N_2 J d\xi & 0 & \int_{-1}^{+1} N_1 N_3 J d\xi \\ \int_{-1}^{+1} N_2 N_1 J d\xi & 0 & \int_{-1}^{+1} N_2 N_2 J d\xi & 0 & \int_{-1}^{+1} N_2 N_3 J d\xi & 0 \\ 0 & \int_{-1}^{+1} N_2 N_1 J d\xi & 0 & \int_{-1}^{+1} N_2 N_2 J d\xi & 0 & \int_{-1}^{+1} N_2 N_3 J d\xi \\ \int_{-1}^{+1} N_3 N_1 J d\xi & 0 & \int_{-1}^{+1} N_3 N_2 J d\xi & 0 & \int_{-1}^{+1} N_3 N_3 J d\xi & 0 \\ 0 & \int_{-1}^{+1} N_3 N_1 J d\xi & 0 & 0\int_{-1}^{+1} N_3 N_2 J d\xi & 0 & \int_{-1}^{+1} N_3 N_3 J d\xi \end{bmatrix} \tag{26a}$$

$$F^{eT} = \left\{ (F_x)_1 (F_y)_1 (F_x)_2 (F_y)_2 (F_x)_3 (F_y)_3 \right\} \tag{26b}$$

$$t^{eT} = \left\{ (t_x)_1 (t_y)_1 (t_x)_2 (t_y)_2 (t_x)_3 (t_y)_3 \right\} \tag{26c}$$

where $F_x$ and $F_y$ are the nodal forces in the $x$ and $y$ directions, respectively. Using the principle of total potential energy minimization, the equivalent stiffness matrix is defined as follows:

$$[K]^{BE} = [M][G]^{-1}[H] \text{ and } \{F\}^{BE} = \int_{\Gamma^{BE}} [N]^T[N]\{t_n\}d\Gamma \tag{27}$$

$$[K]^{BE}\{u_n\}^{BE} = \{F\}^{BE} \tag{28}$$

where $[K]^{BE}$ is the equivalent rigidity matrix of the super-element $BE$ and $\{F\}^{BE}$ its equivalent nodal forces.

The total boundary of the model in the BEM and FEM domains is now divided into two parts: the interaction boundary and the remaining parts of the domain's boundary, as expressed in Equations (29) and (30). Subsequently, the global stiffness matrix can be formulated as follows:

$$\begin{bmatrix} K^{FF} & K^{FI} \\ K^{IF} & K^{II} \end{bmatrix} \begin{Bmatrix} u^F \\ u^I \end{Bmatrix} = \begin{Bmatrix} F^F \\ F^{IF} \end{Bmatrix} \tag{29}$$

$$\begin{bmatrix} K^{BB} & K^{BI} \\ K^{IB} & K^{II} \end{bmatrix} \begin{Bmatrix} u^B \\ u^I \end{Bmatrix} = \begin{Bmatrix} F^B \\ F^{IB} \end{Bmatrix} \tag{30}$$

$$\begin{bmatrix} K^{BB} & K^{BI} & 0 \\ K^{IB} & K^{I} & K^{FI} \\ 0 & K^{IF} & K^{FF} \end{bmatrix} \begin{Bmatrix} u^B \\ u^I \\ u^F \end{Bmatrix} = \begin{Bmatrix} F^B \\ F^{IB} + F^{IF} \\ F^F \end{Bmatrix} \tag{31}$$

where superscripts *F*, *B*, and *I* represent finite element, boundary element, and interface, respectively.

It is crucial to emphasize that, within the framework of the proposed methodology, the BEM is formulated based on elastodynamic equations. This approach is particularly effective for structures exhibiting predominantly linear behavior. Therefore, it is important to acknowledge that the applicability of the proposed method becomes restricted when addressing problems characterized by soil material nonlinearity.

## 5. Evaluation of the Proposed Approach

In this research, a novel code is formulated to analyze the behavior of SSSI. The proposed method is compared with the conventional FEM in solving a specific problem. The results of the comparison demonstrate that the proposed method is highly accurate and acceptable for analyzing SSSI. Furthermore, the developed code can be extended to investigate other types of soil structure interaction problems. The proposed method has the potential to improve the efficiency and accuracy of structural analysis, which can lead to safer and more cost-effective designs.

In order to guarantee the precision and effectiveness of the proposed scheme, two structures have been analyzed using the FEM and the proposed method. The results obtained from the proposed method are compared with those obtained from the FEM, and a comparison of their respective computational efforts is also presented.

### 5.1. Numerical Example Definition for Comparative Analysis

The problem involves two structures placed at a distance from each other, each possessing identical characteristics. These structures are assumed to be made of reinforced concrete, comprising single-span and one-story frameworks. The span length is 6 m, while the height of the structures measures 3 m. Both the columns and beams are presumed to be constructed using rectangular reinforced concrete with dimensions of 0.4 m by 0.4 m. Two specific loadings are applied in this problem, which will be elaborated upon subsequently.

It is assumed that the concrete used in the structures has an elastic modulus of 35 gigapascals, a Poisson's ratio of 0.2, and a mass density of 2500 kg per cubic meter.

The analysis of the soil model in this study involves a rectangular domain situated in two-dimensional space, characterized by assumed elastic plain strain behavior. The dimensions of the soil model, uniformly set at 300 m in width and 30 m in depth, are specifically chosen to accommodate simulations using both the BEM and FEM. This standardized configuration, tailored for structures with a 6 m span, is strategically selected to minimize the adverse effects of soil reflection and refraction at the boundaries.

In addition to the geometric specifications, the soil is presumed to possess distinctive mechanical properties. These include a shearing velocity of 687 m per second, a shear modulus of 0.85 gigapascals, a Poisson's ratio of 0.3, and a mass density estimated at 1800 kg per cubic meter. The comprehensive nature of our study places a significant emphasis on adopting a unified approach for both FEM and the Hybrid Boundary Element—Finite

Element Method. The paramount goal of ensuring consistency across these methodologies is to minimize the influence of extraneous factors, contributing substantially to the overall robustness and reliability of our research outcomes.

The decision to discretize the regions under investigation aligns directly with the specific objectives of our research, which extensively focuses on capturing the intricate dynamics of structure–soil-structure interaction. This approach allows for a more thorough exploration of the complexities involved in understanding and modeling the dynamic interplay between structures and the surrounding soil.

In this problem, the structures are positioned at a distance of 6 m from each other. The first structure is subjected to a force, and the displacement of the first node in the first structure is observed. Figure 3 provides a graphical representation illustrating the schematic model under investigation.

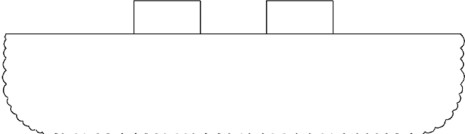

**Figure 3.** Schematic representation of the first SSSI problem model.

Two external loads are exerted on the model under consideration. Figure 4 illustrates a schematic representation depicting the first applied load, known as the Heaviside load. The load in the figure is scaled to a unit load and further modified by the loading factor.

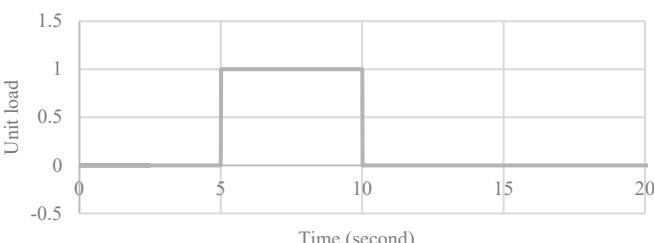

**Figure 4.** The initial or primary applied load for the evaluation problem.

Figure 5 showcases the depiction of the second load, appropriately scaled to a value of 1. This load is constructed through the sequential application of two step functions, with the second step function being activated 5 s after the initiation of the first step function.

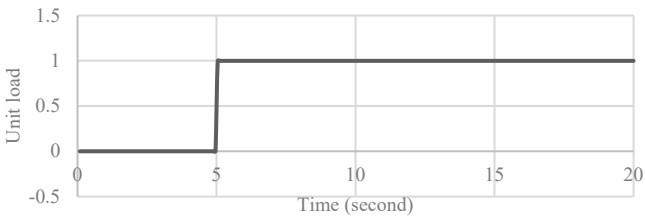

**Figure 5.** The subsequent or secondary applied load for the evaluation problem.

### 5.2. Finite Element Model: Structural Analysis and Results

The finite element model is constructed using the ABAQUS 2021 software package, renowned for its robust capabilities in structural analysis. This model consists of three distinct components: the first and second parts represent the structures, while the third part represents the soil element.

The modeling of each structure involves the utilization of 12 linear line elements of type B21, enabling an accurate representation of their geometrical and mechanical properties. In parallel, the soil domain is discretized using 1000 quadratic quadrilateral elements of type CPE8REL, allowing for a refined analysis of the soil's behavior. The combined domain

encompasses a total of 1024 elements, providing a comprehensive representation of the entire system.

In order to maintain a coherent and consistent modeling approach, it is postulated that the damping characteristics within the domain align with the damping principles employed in the BEM, specifically the Rayleigh damping method [41]. By adopting this assumption, the damping coefficients utilized in both the BEM and the current model are treated as identical. This congruity in damping coefficients ensures the preservation of system integrity and enables a unified treatment of damping effects throughout the entirety of the analysis process.

In order to establish contact between the structures and the soil domain, a hierarchical arrangement is employed, wherein the soil body is designated as the master part, while the structures are regarded as slave parts within the model. At the interface between the soil and the structures, the degrees of freedom pertaining to the contact points are tightly coupled, ensuring that any displacement or motion in one entity is faithfully transmitted to the other. Additionally, it is postulated that the rotational degrees of freedom in both structures remain unrestricted, allowing for free rotation without constraint.

The elements employed within the FEM are depicted in Figure 6. This illustration exhibits the soil domain positioned beneath the structures, characterized by quadratic plain strain elements, while the structures themselves are represented by linear beam elements. The coordination of these domains is focused on the spatial region between the two structures, and the coordinate center is also displayed in Figure 6.

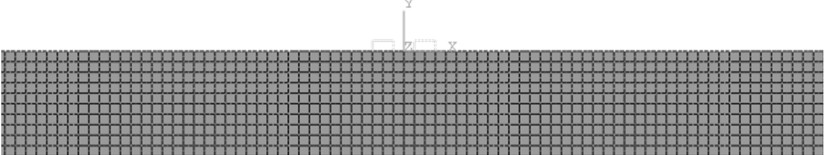

**Figure 6.** Elements utilized in the fem model for the initial problem.

### 5.3. The Proposed Method (FEM/BEM): Structural Analysis and Results

In the proposed coupled finite element-boundary element method (FEM–BEM) approach, the body of the soil is modeled using the BEM, while the structures are modeled using the FEM. The soil body is discretized into a total of 66 elements, while each structure is represented by 4 beam elements. Thus, the overall model in the proposed approach consists of 74 elements. Figure 7 visually illustrates the model created by the proposed approach, highlighting the reduced number of elements compared to other methods and resulting in decreased computational effort.

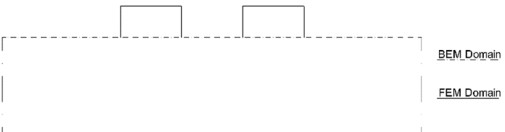

**Figure 7.** A schematic illustration of the constituent elements encompassed in the model developed through the proposed method for the evaluation problem.

Furthermore, Figure 7 demonstrates that the density of elements is higher at the top of the soil compared to the sides and base of the soil. The structures, similar to the FEM model, are centrally positioned within the model and are located adjacent to each other with a spacing of 6 m.

### 5.4. Comparison of the Results in Transient Domain

The comparative evaluation of the results for the initial problem in the transient domain reveals a notable alignment between the results obtained through both approaches. The proposed method demonstrates a reduced computational effort without compromising

the accuracy compared to the FEM. Conversely, the FEM exhibits a slower convergence rate in comparison. Figure 8 illustrates the outcome of the first loading for the initial problem.

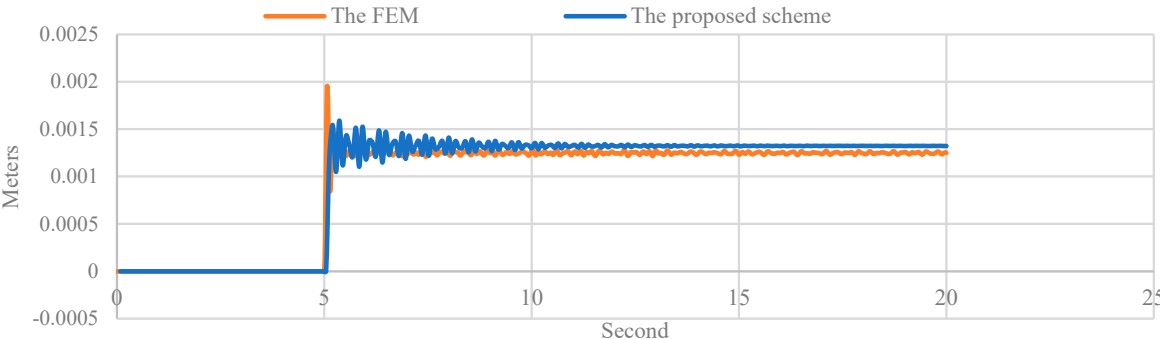

**Figure 8.** Displacement comparison of the first node in the first structure during initial loading.

Similar to the initial loading phase, a significant similarity is observed in the displacement observed at the first node of the primary structure between the proposed method and the FEM. This parallelism becomes evident when comparing the outcomes obtained from both approaches. To visually represent this comparison, Figure 9 presents the results for the second loading scenario within the context of the evaluation problem.

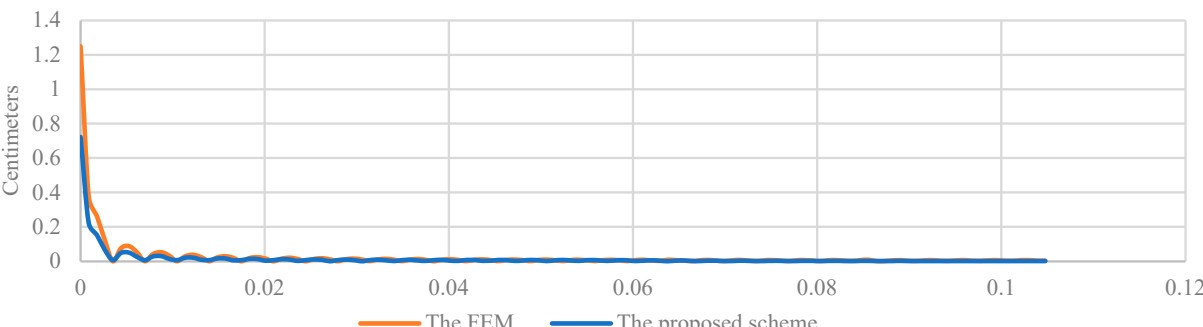

**Figure 9.** Displacement comparison of the first node in the first structure during second loading.

### 5.4.1. Dimensionless Comparison

In order to facilitate a dimensionless comparison of results, the dimensionless frequency of a0 is introduced. This frequency is obtained by dividing the angular frequency, expressed in hertz, by pi, and the velocity of the soil, and then multiplying it by the inter-column spacing within the structure. The calculation can be expressed as follows:

$$a0 = \omega b / \pi c \tag{32}$$

The variable $a0$ is used to represent the dimensionless frequency, where $\omega$ denotes the angular frequency, $b$ indicates the span of the frames, and $c$ represents the shearing wave velocity of the soil. In light of this, Figure 10 illustrates the displacement results obtained for the first node of the primary structure during the initial loading phase of the evaluation problem.

The alteration of the loading conditions has the potential to impact the behavioral pattern of the structures. Consequently, the same principle is applied to examine the response of the structure under the second loading scenario. To elucidate this comparison, Figure 11 presents the displacement results obtained for the first node of the primary structure during the second loading phase of the evaluation problem.

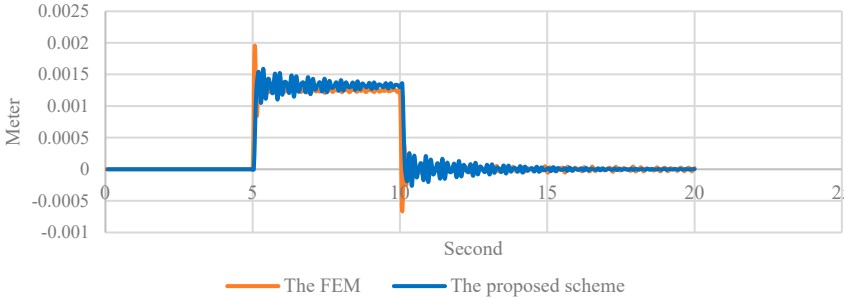

**Figure 10.** Displacement comparison of the first node in the first structure during initial loading in unitless domain.

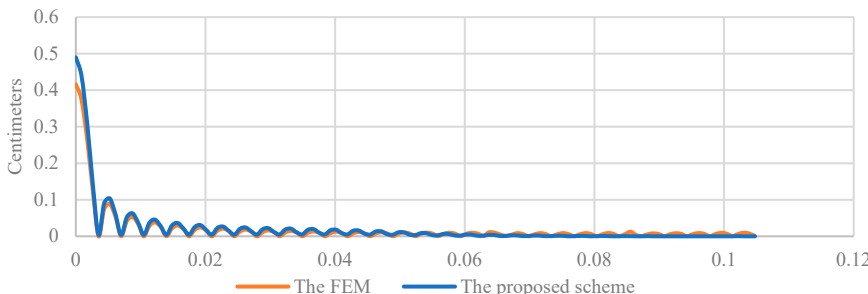

**Figure 11.** Displacement comparison of the first node in the first structure during second loading in unitless domain.

### 5.4.2. Result of the Deformation on the Soil

The influence of soil deformation holds considerable importance in shaping the behavior of the system. This study investigates a proposed coupling method to evaluate the impact of soil deformation on the surface. Figure 12 exhibits the deformation outcomes of the soil at different distances from the structure's center for the initial load pattern outlined in Figure 3, at a frequency of 1 hertz, as indicated in the figure. With the exception of the regions adjacent to the boundaries, the observed values of soil deformation are deemed reasonable and fall within acceptable thresholds.

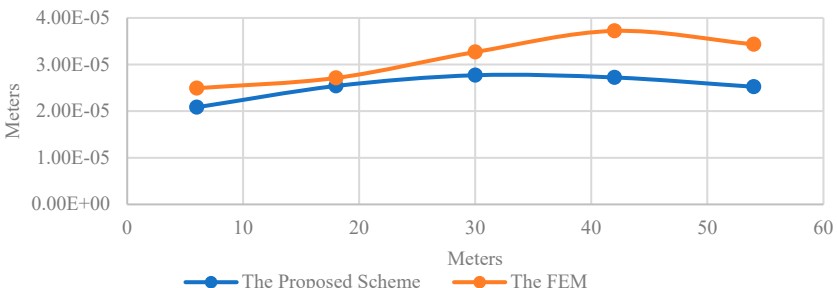

**Figure 12.** Deformation results of the soil at various distances from the center of the structure during the first loading phase of the problem in 1 hertz frequency.

### 5.4.3. Comparison of the Computational Effort

The computational effort of the proposed scheme is evaluated in comparison to that of the FEM. The initial analysis indicates that the proposed method exhibits reduced computational effort compared to the FEM, resulting in faster convergence times. However, it should be noted that the proposed scheme, despite addressing certain limitations of the BEM, experiences a significant increase in computational effort with an increase in the number of time steps. Therefore, it can be concluded that the effectiveness of the proposed method is particularly notable for a lower number of time steps, while the benefits may diminish as the number of time steps increases. Table 2 provides a comparative analysis

of the computational effort between the two methods, while Table 3 visually depicts the progressive increase in computation time associated with the proposed method.

**Table 2.** Comparative analysis of computational effort: proposed scheme vs. FEM.

| Item | Method | Problem | Computational Effort (s) |
|:---:|:---:|:---:|:---:|
| 1 | Proposed method | 1 | 276 |
| 2 | FEM | 1 | 1201 |
| 3 | Proposed method | 2 | 269 |
| 4 | FEM | 2 | 1200 |

**Table 3.** Computational effort for initial and final time steps in two 500-step analyses.

| Item | Time Step | Problem | Computational Effort (s) |
|:---:|:---:|:---:|:---:|
| 1 | First | 1 | 0.46 |
| 2 | Last | 1 | 0.75 |
| 3 | First | 2 | 0.45 |
| 4 | Last | 2 | 0.80 |

## 6. Numerical Analysis of the Influence of Inter-Structure Spacing on SSSI Magnitude

The distance between two structures is recognized as a pivotal factor influencing the magnitude of structure–soil-structure interaction, constituting a crucial aspect of analysis [42]. In the forthcoming section, an examination is conducted through numerical analysis to investigate the influence of inter-structure spacing on the magnitude of soil–structure interaction.

The model employed for investigating the effect of inter-structure spacing is a two-dimensional model. In this model, the soil is simulated using a two-dimensional domain of elastic plane strain with dimensions of 300 m by 30 m. In the realm of structural modeling, the utilization of two-dimensional static elastic beam elements has been adopted. The modeling framework encompasses two distinct structures, each possessing unique characteristics. The initial structure is characterized as a single-story entity, exhibiting a vertical extent of 3 m and a horizontal span of 6 m. In contrast, the second structure is designed as a four-story construct, wherein each floor features a height of 3 m, while maintaining a consistent span of 6 m between successive floors. These structures are positioned atop the ground surface in close proximity to one another.

To comprehensively explore the influence of inter-structure spacing, a total of nine models were investigated in this study. To achieve this, three different types of soil were employed, along with three distinct inter-structure spacing configurations. In the initial model, the distance between the structures was assumed to be 0.5 m, while the second model involved a spacing of 1 m. Finally, the last model considered a larger separation between the structures, with a distance of 2 m. This systematic variation in inter-structure spacing allowed for a comprehensive examination of the effects and trends associated with different distances on the studied parameters.

The construction of these models involved the use of C30 concrete, exhibiting a specific compressive strength of 30 megapascals and an elastic modulus of 30 gigapascals. The concrete, inclusive of embedded reinforcement, was assumed to possess an average unit weight of 2500 kg per cubic meter. Three distinct soil types were employed, distinguished by their respective shear wave velocities. The first soil type corresponds to soft soil, characterized by a shear wave velocity of 471.40 m per second. The second soil type represents medium soil, exhibiting a shear wave velocity of 666.67 m per second. Lastly, the third soil type is classified as stiff soil, with a shear wave velocity of 1333.33 m per second. The elastic properties associated with these soil types are detailed in Table 4. Figure 13 presents schematic representations of the three aforementioned model types.

**Table 4.** Elastic properties of the three assumed soils beneath the structures.

| Specification | Unit | Stiff Soil | Medium Soil | Soft Soil |
|---|---|---|---|---|
| Shear Modulus G | N/m$^2$ | $3.2 \times 10^9$ | $8 \times 10^8$ | $4 \times 10^8$ |
| Young's Modulus E | N/m$^2$ | $8 \times 10^9$ | $2 \times 10^9$ | $1 \times 10^9$ |
| Specific Mass ρ | Kg/m$^3$ | 1800 | 1800 | 1800 |
| Poisson's Ratio ν | - | 0.25 | 0.25 | 0.25 |
| Shear Wave Velocity Cs | m/s | 1333.33 | 666.67 | 471.40 |

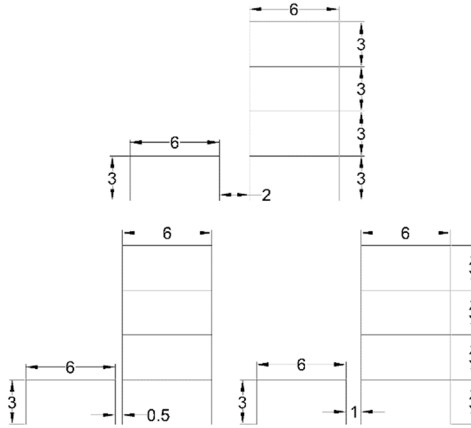

**Figure 13.** Models of the structures with varying inter-structure distances.

The procedure of analysis is as follows: The initial structure assumes the burden of the externally applied load, localized specifically on the leftward region of its roof. Displacement measurements are obtained from the roofs of both the first and second structures. A graphical representation, denoted as Figure 14, showcases the load distribution pattern. Subsequently, this distribution pattern is subjected to amplification by a multiplicative factor of 10,000, and subsequently it is imposed upon the roof of the first structure. The duration of this load application persists for a period of one second, and the overall analysis encompasses a time span of ten seconds, encompassing the precise segment in which the load is applied.

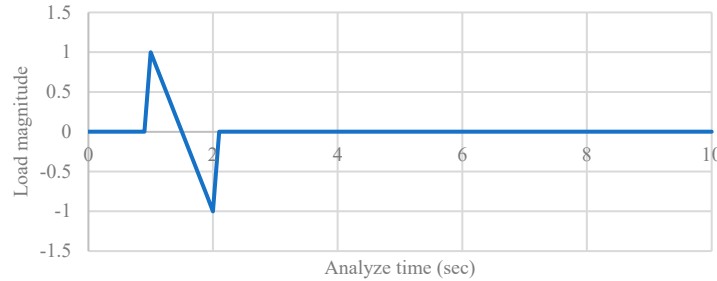

**Figure 14.** Loading configuration throughout the analysis duration.

### 6.1. The Effects of Dynamic Wave Propagation between Two Structures in Hard Soil

The shear wave velocity exhibits a significant influence on the dynamic response of soil. In this investigation, the hard soil is presumed to possess a shear wave velocity of 1333.33 m per second. The inter-structural spacing encompasses a range from 0.5 m to 2 m, and the consequential deformation of both the first and second structures, moving from left to right, is quantified on the roof of said structures. The measured deformations are presented both in the time domain and in the dimensionless domain denoted as $a_0$. As depicted in Figure 15, a comparative analysis of roof deformation is presented specifically for the first structure.

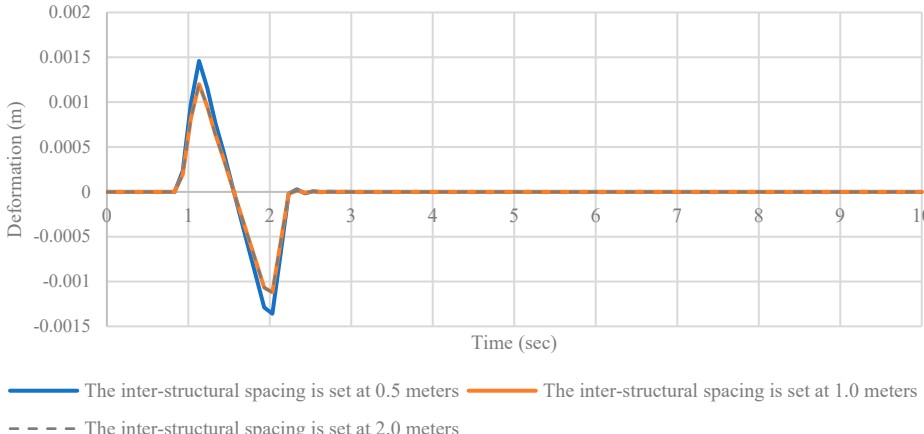

**Figure 15.** Deformation patterns over time with varied inter-structural spacings in the first structure.

The results derived from the deformation analysis performed on the first structure reveal that the inter-structural spacing exerts an influence on the deformation experienced by the first structure, albeit with a relatively modest degree of modification. This effect can be attributed to two possible factors: the impact of the second structure on the soil's stiffness or the inertial wave generated by the second structure, affecting the first structure. Figure 16 illustrates the impact of the second structure on the deformation characteristics of the first structure, specifically observed within the dimensionless domain denoted as $a_0$.

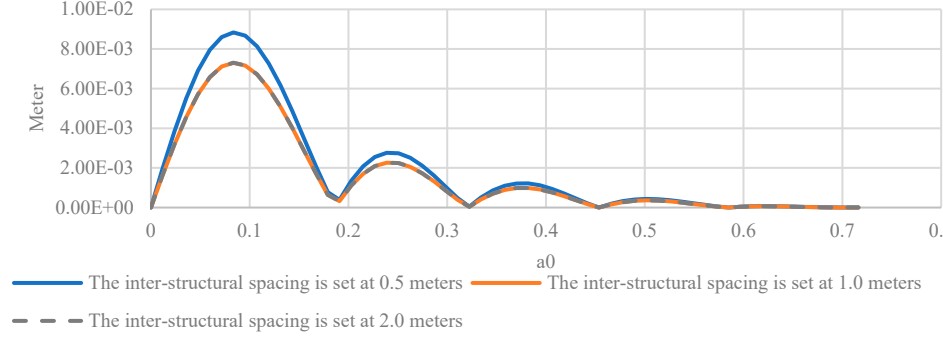

**Figure 16.** Influence of the second structure on deformation characteristics of the first structure in the dimensionless domain ($a_0$).

The subsequent analysis explores the deformation patterns observed in the second structure under different inter-structural spacings. Similar to the first structure, the deformation results are presented both in the time domain and in the dimensionless domain represented by $a0$. Figure 17 depicts a comparative analysis of deformation patterns within the time domain for the roof of the second structure, considering various inter-structural spacings.

The outcomes derived from the analyses presented in Figure 18 demonstrate a significant decrease in the influence of the first structure as the inter-structural distance between the two structures increases. Figure 17 illustrates the impact of progressively increasing the inter-structural distance between two structures on the deformation of the roof in the second structure, considering a hard soil condition in unit less domain of $a_0$.

As evidenced in Figures 15–18, augmenting the inter-structural distance between structures subjected to external loading significantly impacts the deformation of the recipient structure that receives the wave from the loaded structure. Conversely, the structural element on which the load is applied exhibits a relatively minor response to the increasing inter-structural distance.

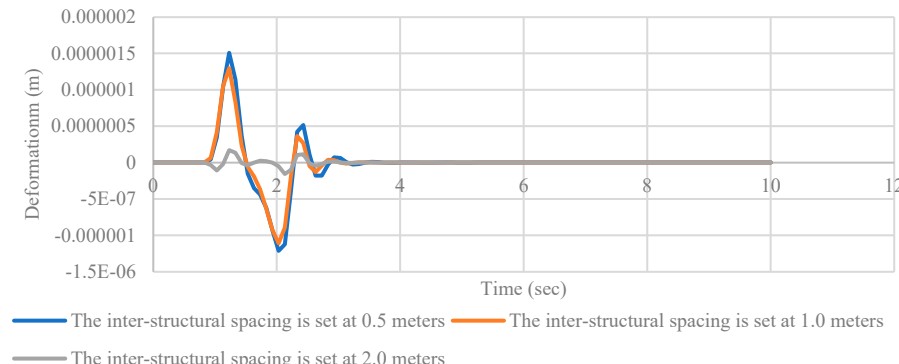

**Figure 17.** Comparative analysis of deformation patterns over time with varied inter-structural spacings for the second structure.

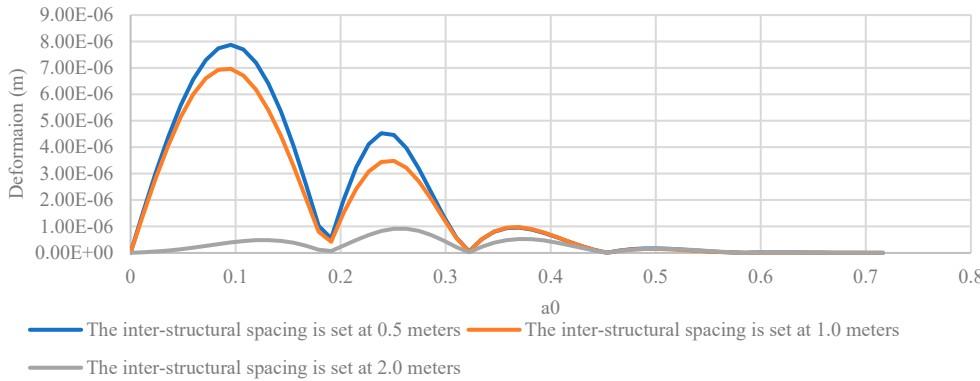

**Figure 18.** Comparison of deformation patterns within the $a_0$ domain across varied inter-structural spacings for the second structure.

### 6.2. The Effects of Dynamic Wave Propagation between Two Structures in Medium Soil

This investigation assumes a shear wave velocity of 666.67 m per second for the medium soil. The inter-structural spacing ranges from 0.5 m to 2 m, and the resulting deformation of both structures is quantified on their respective roofs, progressing from left to right. The measured deformations are presented in both the time domain and the dimensionless domain ($a_0$). Figure 19 provides a comparative analysis of roof deformation specifically for the first structure in medium soil.

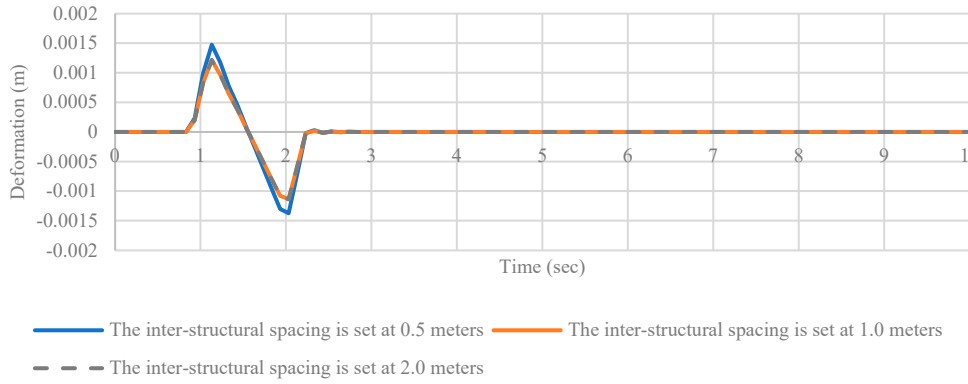

**Figure 19.** Detailed comparative analysis of roof deformation patterns in the first structure under medium soil conditions.

The findings of the deformation analysis conducted on the first structure reveal that, even in a medium soil condition, the inter-structural spacing exerts a noticeable but moderate influence on its deformation of the loaded structure. Figure 20 graphically illustrates

the effect of the second structure on the deformation characteristics of the first structure, specifically within the dimensionless domain represented as $a_0$.

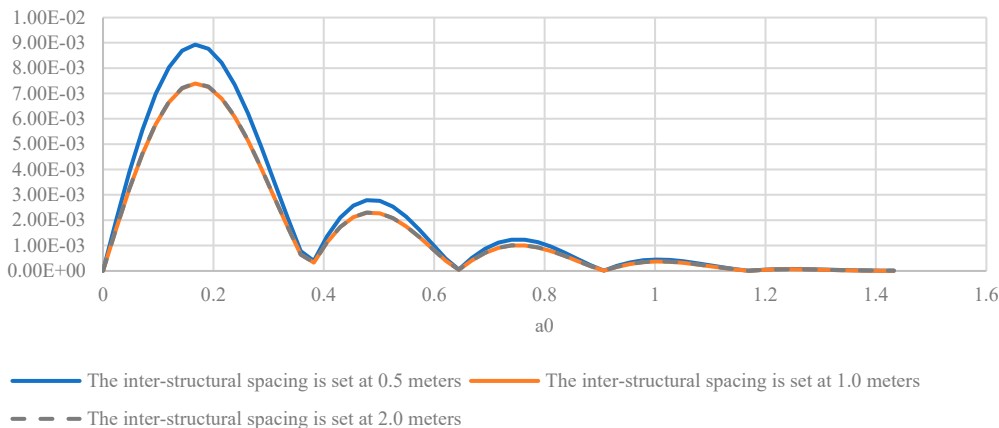

**Figure 20.** Impact of the second structure on deformation characteristics of the first structure in the dimensionless domain ($a_0$) under medium soil conditions.

The subsequent analysis investigates the deformation patterns exhibited by the second structure across varying inter-structural spacings. Similar to the first structure, the deformation results are presented in both the time domain and the dimensionless domain, represented by $a_0$. Figure 21 portrays a comparative analysis of deformation patterns within the time domain specifically for the roof of the second structure, encompassing a range of inter-structural spacings.

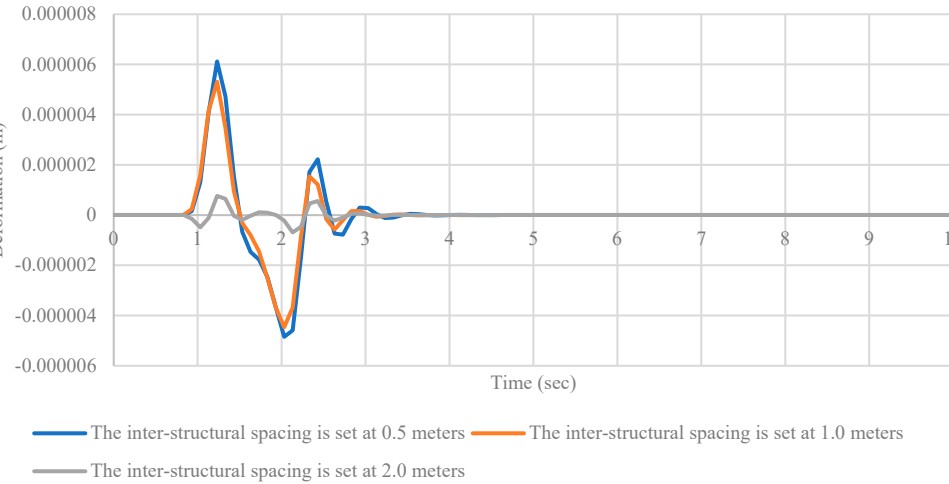

**Figure 21.** Comparative assessment of deformation patterns over time with varied inter-structural spacings in the second structure under medium soil conditions.

The analysis presented in Figure 21, like the results presented in Figure 17, demonstrates a significant decrease in the influence of the first structure as the inter-structural distance between the two structures increases. Figure 22 provides a visual representation of the impact of progressively increasing the inter-structural distance between two structures on the deformation of the roof in the second structure. This analysis considers the presence of a hard soil condition and is evaluated within the dimensionless domain of $a_0$ in medium soil.

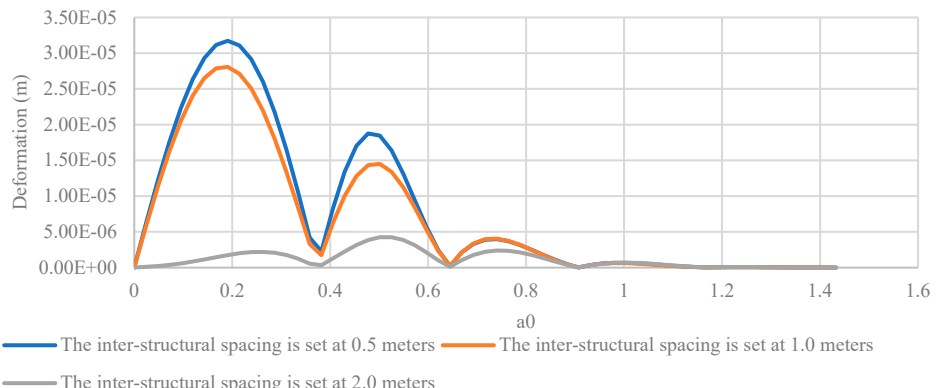

**Figure 22.** Comparative assessment of deformation patterns in the $a_0$ domain with varied inter-structural spacings in the second structure in a medium soil environment.

The analysis depicted in Figures 19–22 demonstrates that augmenting the inter-structural distance between loaded structures has a significant impact on the deformation of the receiving structure influenced by the wave from the loaded structure. In contrast, the structure on which the load is applied exhibits a relatively minor response to the expanding inter-structural distance, exhibiting a behavior pattern similar to that observed in hard soil conditions.

### 6.3. The Effects of Dynamic Wave Propagation between Two Structures in Soft Soil

This section presupposes a shear wave velocity of 471.40 m per second for the soft soil. The distance between structures like other soils varies between 0.5 m and 2 m, and the resultant deformation of each structure is evaluated on their respective rooftops, proceeding from left to right. The recorded deformations are reported in both the temporal domain and the dimensionless domain ($a_0$). Figure 23 presents a comprehensive examination of roof deformation specifically for the initial structure situated in soft soil.

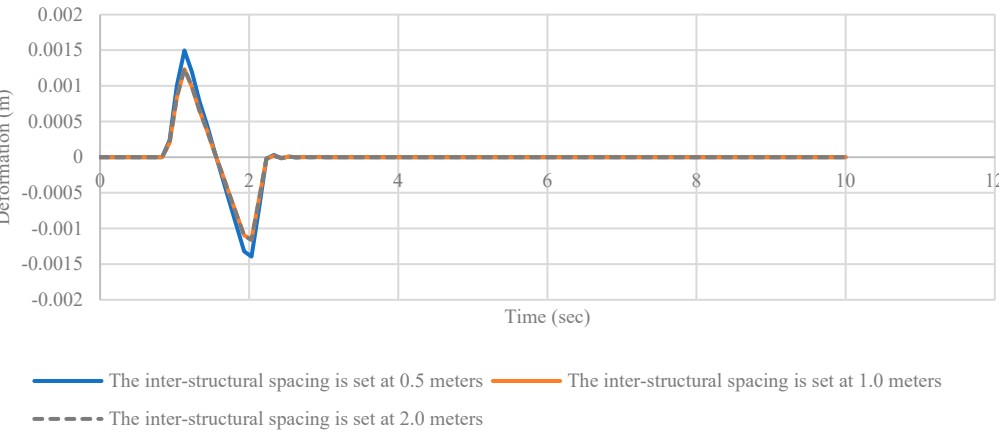

**Figure 23.** Comparative assessment of deformation patterns over time with varied inter-structural spacings in the first structure under soft soil conditions.

The results derived from the deformation analysis performed on the initial structure demonstrate that, despite the soft soil condition, the inter-structural spacing exhibits a discernible yet moderate impact on the deformation behavior of the loaded structure. To visually portray the influence of the second structure on the deformation characteristics of the first structure, particularly in the dimensionless domain denoted as $a_0$, Figure 24 provides a graphical representation.

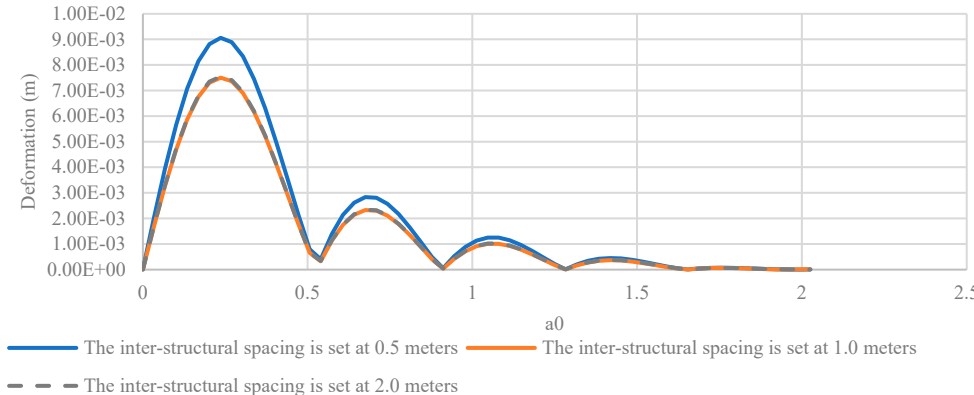

**Figure 24.** Influence of the second structure on deformation characteristics of the first structure in the dimensionless domain ($a_0$) under soft soil conditions.

The present investigation examines the deformation patterns displayed by the second structure under different inter-structural spacings. Similar to the analysis conducted on the first structure, the deformation outcomes are showcased in both the temporal domain and the dimensionless domain, denoted by $a_0$. Figure 25 depicts a comparative assessment of deformation patterns in the temporal domain, specifically focusing on the roof of the second structure, encompassing a variety of inter-structural spacings.

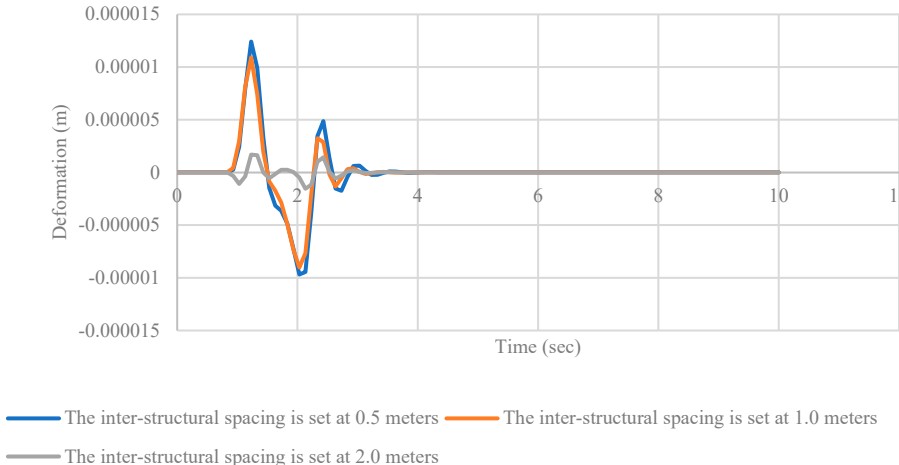

**Figure 25.** Comparative assessment of deformation patterns in the temporal domain with varied inter-structural spacings in the second structure, emphasizing the influence of soft soil conditions.

The investigation illustrated in Figure 25, similar to the observations presented in Figures 17–21, demonstrates a notable decrease in the influence exerted by the first structure as the inter-structural distance between the two structures increases. Figure 26 offers a visual depiction of the influence exerted by incrementally increasing the inter-structural distance between the two structures on the deformation of the roof in the second structure. This analysis takes into account the existence of a rigid soil condition, and is assessed within the dimensionless domain of $a_0$ in soft soil.

The analysis presented in Figures 23–26 reveals a noteworthy effect of increasing the inter-structural distance between loaded structures on the deformation of the receiving structure under the influence of the wave propagated from the loaded structure. Conversely, the structure subjected to the applied load demonstrates a comparatively minor response to the widening inter-structural distance, displaying a behavior reminiscent of that observed in rigid soil conditions. This phenomenon is applicable across various soil types, and is independent of the soil type upon which the structure is built.

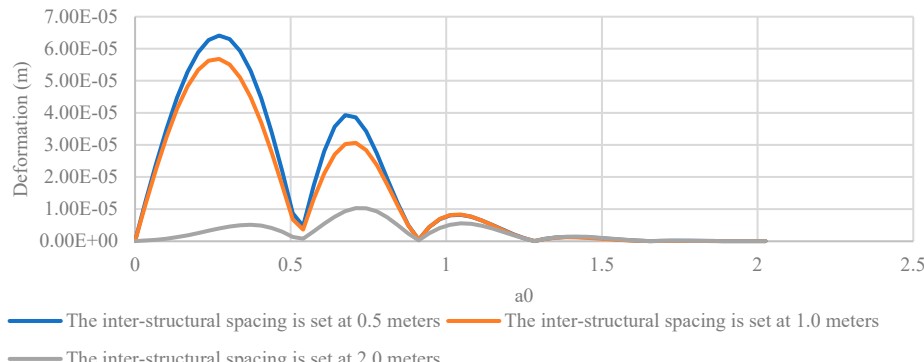

**Figure 26.** Comparative assessment of deformation patterns in the $a_0$ domain for the second structure, considering diverse inter-structural spacings in a soft soil setting.

## 7. Conclusions

This research paper introduces an innovative technique for the numerical simulation of Structure–Soil-Structure Interaction (SSSI) through the synergistic integration of the Boundary Element Method (BEM) and Finite Element Method (FEM) within a coupled framework. The study focuses on conducting a numerical analysis to investigate the influence of inter-structural distance on the phenomenon of soil–structure interaction. The proposed methodology is subjected to rigorous accuracy assessment, employing a comprehensive comparative analysis against the results obtained solely from the FEM method. Furthermore, the computational efficiency of the coupled BEM–FEM approach is thoroughly evaluated and compared to that of the standalone FEM method.

The comparative analysis presented in this study highlights the superior capabilities of the proposed methodology, thereby underscoring its potential as a highly effective and efficient approach for addressing challenges related to SSSI encountered in engineering applications. The effectiveness of the proposed method is demonstrated by its capacity to yield precise results while concurrently alleviating computational burden, as evidenced by the reduced convergence time compared to the FEM. However, it is important to acknowledge that the computational effort required by the proposed method increases as the number of time steps increases. Moreover, the results obtained from the performed analyses demonstrate that the augmentation in the inter-structural distance, especially in scenarios involving dynamic loading, exerts a substantial influence on the behavior of the non-loaded structure, regardless of the soil type upon which the structure is constructed. In contrast, the impact on the loaded structure is comparatively minor.

In conclusion, this study makes a significant contribution to the progression of numerical modeling methodologies in the realm of SSSI, providing a valuable technique that resonates with both engineers and researchers in this field. The proposed method's capacity to deliver accurate and efficient outcomes, while effectively addressing the intricate challenges associated with SSSI, renders it a promising tool for a wide array of engineering applications that involve the interaction between structures and soil. Furthermore, the presented results of the numerical examples aim to provide engineers with valuable insights into the magnitude and nature of the effects that SSSI can exert on the structural behavior.

**Author Contributions:** Conceptualization, P.A., J.A.M. and M.P.; methodology, P.A.; software, P.A.; validation, P.A., J.A.M. and M.P.; resources, P.A., J.A.M., M.P. and M.S.R.; data curation, P.A.; writing—original draft preparation, P.A.; writing—review and editing, P.A., J.A.M. and M.P.; visualization, P.A.; supervision, J.A.M., M.P. and M.S.R.; project administration, J.A.M. All authors have read and agreed to the published version of the manuscript.

**Funding:** This research received no external funding.

**Data Availability Statement:** The data presented in this study are available on request from the corresponding authors.

**Conflicts of Interest:** The authors declare no conflicts of interest.

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
