# Peer review of "A Coupled Finite-Boundary Element Method for Efficient Dynamic Structure-Soil-Structure Interaction Modeling"

_mca, doi:10.3390/mca29020024_

Round 1

Reviewer 1 Report

Comments and Suggestions for Authors

The detailed comments can be found from the attached PDF file. 

Comments on the Quality of English Language

The English still can be improved further by clearing the spelling and grammer mistakes. Some have been pointed in the comments which can be found from the attached PDF file, but the authors still should check the manuscript throughly. 

Author Response

First of all, the authors really appreciate the editor and the reviewers for their recognition and suggestions. According to the suggestions of the editor and the reviewers, the authors have checked and revised the manuscript carefully. Please find the attached response file.

Reviewer 2 Report

Comments and Suggestions for Authors

The paper entitled “A Coupled Finite-Boundary Element Method for Efficient Dynamic Structure-Soil-Structure Interaction Modeling” introduces an approach to numerically model Structure-Soil-Structure Interaction (SSSI) by integrating the Boundary Element Method (BEM) and the Finite Element Method (FEM) in a coupled manner. The proposed approach provides more reliable results compared to the traditional FEM method.

COMMENTS

Figure 2.:

3rd Box:  Finite Element Method Formulation

4th Box:  Boundary Element Method Formulation

[NOT Elements]

Figure 2.: For A, X, F use:  [A], {X}, {F}

LINES 232: 3. Boundary Element Method (BEM)

LINES 233: The Boundary Element Method is a numerical technique ...

[NOT Elements].

LINES 400-403: “Furthermore, Figure 7 demonstrates that the density of elements is higher at the top of the soil compared to the sides and base of the soil.”

WHY YOU DISCRETIZE THE SIDES AND THE BASE OF THE SOIL ?

By proper BEM formulation, we can discretize only the top surface of the soil, and not the “artificial” sides and “artificial” base of the soil. BEM is perfect for semi-infinite domains.  The capability of the BEM to model infinite domains accurately and efficiently, without the need for numerical artifices, makes it the perfect complement to the FEM.

Comments on the Quality of English Language

Minor editing of the English language is required.

Author Response

(The authors gave the same response as above.)

Round 2

Reviewer 2 Report

Comments and Suggestions for Authors

The revised paper can now be accepted.